# The Hamburg Spondylodiscitis Assessment Score (HSAS) for Immediate Evaluation of Mortality Risk on Hospital Admission

**DOI:** 10.3390/jcm11030660

**Published:** 2022-01-27

**Authors:** Annika Heuer, André Strahl, Lennart Viezens, Leon-Gordian Koepke, Martin Stangenberg, Marc Dreimann

**Affiliations:** 1Division of Spine Surgery, Department of Trauma and Orthopedic Surgery, University Medical Center Hamburg-Eppendorf, 20251 Hamburg, Germany; l.viezens@uke.de (L.V.); l.koepke@uke.de (L.-G.K.); m.stangenberg@uke.de (M.S.); m.dreimann@uke.de (M.D.); 2Division of Orthopedics, Department of Trauma and Orthopedic Surgery, University Medical Center Hamburg-Eppendorf, 20251 Hamburg, Germany; a.strahl@uke.de

**Keywords:** spondylodiscitis, spinal infection, risk score, spine surgery, vertebral osteomyelitis, prognostic factors

## Abstract

(1) Background: Patients with spondylodiscitis often present with unspecific and heterogeneous symptoms that delay diagnosis and inevitable therapeutic steps leading to increased mortality rates of up to 27%. A rapid initial triage is essential to identify patients at risk for a complicative disease course. We therefore aimed to develop a risk assessment score using fast available parameters to predict in-hospital mortality of patients admitted with spondylodiscitis. (2) Methods: A retrospective data analysis of 307 patients with spondylodiscitis recruited from 2013 to 2020 was carried out. Patients were grouped according to all-cause mortality. Via logistic regression, individual patient and clinical characteristics predictive of mortality were identified. A weighted sum score to estimate a patient’s risk of mortality was developed and validated in a randomly selected subgroup of spondylodiscitis patients. (3) Results: 14% of patients with spondylodiscitis died during their in-hospital stay at a tertiary center for spinal surgery. Univariate and logistic regression analyses of parameters recorded at hospital admission showed that age older than 72.5 years, rheumatoid arthritis, creatinine > 1.29 mg/dL and CRP > 140.5 mg/L increased the risk of mortality 3.9-fold, 9.4-fold, 4.3-fold and 4.1-fold, respectively. *S. aureus* detection increased the risk of mortality by 2.3-fold. (4) Conclusions: The novel Hamburg Spondylodiscitis Assessment Score (HSAS) shows a good fit identifying patients at low-, moderate-, high- and very high risk for in hospital mortality on admission (AUC: 0.795; *p* < 0.001). The implementation of the HSAS into clinical practice could ease identification of high-risk patients using readily available parameters alone, improving the patient’s safety and outcome.

## 1. Introduction

Over the last decade the annual incidence rate of spondylodiscitis (S) has risen dramatically by around 150–260%, leading to lengthy hospital stays and long-term socioeconomic burden [1,2,3,4]. The majority of patients diagnosed with S report a wide array of symptoms including back and/or neck pain, with or without fever [5,6]. Patients commonly present after several weeks or months of progressively worsening symptoms, thus delaying the initiation of early treatment. When suspected, every effort should be made to enable rapid diagnostics starting with imaging, routine laboratory testing and identification of the pathogen by blood culture and/or biopsy of the infected intervertebral space or vertebral bone [6,7]. Without accompanying neurological deficits, the prognosis is improved if antibiotic therapy and surgical management, if necessary, are initiated at an early stage [8]. Despite advances in diagnostic modalities and increasing medical awareness, S is frequently overlooked, misdiagnosed or mismanaged due to non-specific symptoms. Unfortunately, delayed diagnosis of S correlates with unfavorable treatment outcomes as well as a mortality rate of up to 27% [9,10,11,12].

Known risk factors associated with an increased mortality of S include advanced age, intravenous drug abuse, infective endocarditis, degenerative spine disease, prior (spinal) surgery, liver cirrhosis, malignancies, hemodialysis, diabetes mellitus, corticosteroid therapy and other immunocompromised states [2,7,11,13]. Despite adequate treatment, sequelae resulting from chronic pain and persistent disability are high, with less than one-third of all patients achieving full recovery [5,7,10].

Early identification of high-risk patients could improve patients’ safety and care [14,15,16]. We therefore sought to develop a risk score, combining individual patient characteristics and laboratory findings easily accessible at the time of admission, to allow a rapid prediction of individual risk of mortality in a tertiary center for spinal surgery and infectious diseases.

## 2. Materials and Methods

### 2.1. Study Design and Ethics Statement

The study was conducted in compliance with ethical standards and acknowledged by the local Ethics committee (WF013-20). Prospectively, demographic and disease-relevant medical data of patients treated due to proven S at our department for spine surgery were collected and anonymously registered in an electronic database. Data sets of 307 consecutive cases between 2013 and 2020 were available for current retrospective analysis.

### 2.2. Diagnosis and Treatment

Diagnosis was based on clinical presentation, radiographic imaging, laboratory parameters and microbiological diagnostic according to clinical practice guidelines [6,17]. In all cases the route of infection was either hematogenous or per continuitatem. Postoperative spinal infections and patients with suspected gout were excluded. As diagnostic gold standard, magnetic resonance imaging (MRI) was used to screen for infected intervertebral discs and adjacent vertebra, as well as epidural abscesses. In disseminated cases at least two spinal regions (cervical, thoracic or lumbar) were affected, and segments were added for analysis. To determine osseous integrity, computed tomography (CT) was performed. The American Spinal Injury Association Scale was utilized to evaluate neurological deficits (AIS score). To screen for endocarditis, all patients routinely underwent transthoracic echocardiography (TTE). When TTE proved inconclusive, transesophageal echocardiography was performed by a board-certified cardiologist [13]. Each patient was discussed at a weekly interdisciplinary infectious disease conference where the course of treatment, duration of intravenous antibiotic treatment or switch to oral antibiotic treatment was determined individually [18]. Screening for concomitant infections and/or the primary infectious source followed a standardized protocol upon admission, which is comparable to previously described concepts [18]. Three pairs of peripheral blood cultures and a sterile urine culture are extracted at the emergency unit and regularly during hospitalization. Additionally, chest X-rays are conducted. Contrast enhancing computer tomography (CT) or positron emission tomography are reserved for individual approaches following interdisciplinary case discussion, i.e., when relevant abdominal foci are suspected or systemic infection persists. Further, a clinical examination accompanied by an orthopantomogram, if needed, was performed by an oral and maxillofacial surgeon to screen for dental or craniofacial foci. Joint punctures were performed in all artificial joints where prior endoprosthesis surgery was performed to screen for implant infection by cell count and microbiological testing. Foreign materials (port/central venous catheter) were changed or removed upon admission and changed regularly when bacteremia persisted.

Surgical treatment was performed if patients presented in septic conditions, with neurological impairment, spinal instability or deformity, presence of large epidural, or paraspinal abscess formation, as well as due to severe bony destruction with or without instability and fracturing [19,20]. Secondary criteria for surgical treatment were uncontrollable pain or failure to respond to antibiotic therapy alone [14,21,22]. Invasive procedures ranged from transpedicular diagnostic biopsy, minimally invasive decompressive surgery alone, intervertebral fusion with posterior stabilization, as well as anterior corpectomy or combined dorso-ventral and ventro-dorsal approaches. The main reason for stabilization was instability due to bony defects (Figure 1 and Figure 2). Thorough debridement of the intervertebral disc space was performed in all patients treated with intervertebral fusion or corpectomy. Furthermore, at least three to five intraoperative tissue biopsies were sampled for microbiological testing via culture and PCR/qPCR. A separate sample was collected for pathohistological analysis. During the course of treatment, routine chemical laboratory measurements were performed.

### 2.3. Statistical Analysis

All analyses were performed using SPSS version 25 (IBM, Armonk, New York, NY, USA) for Windows. Statistical significance was set to a 2-tailed *p*-value of <0.05. Continuous variables are expressed as mean ± standard deviation (SD), while categorical variables are expressed as a number and percentage (%). To compare patients in terms of continuous variables, the Student’s *t*-test for independent samples was used for normally distributed data. The Mann−Whitney U test was used with non-normally distributed data, and chi-squared or Fisher’s exact tests were applied for categorial variables.

To identify potential predictors of mortality, all deceased and surviving patients with S were subjected to regression analysis. A first logistic regression was performed to determine whether age, sex, comorbidities (congestive heart failure, rheumatoid arthritis, chronic kidney disease, endocarditis, diabetes mellitus etc.), microbiological results (blood cultures and/or intraoperative samples) and preoperative routine laboratory parameters predicted the occurrence of mortality within the examined patient population.

For all significant continuous predictors, a threshold was defined by means of Youden’s J statistic. The Youden Index is a measure that uses Receiver Operating Characteristic (ROC) curves to determine the best threshold value to distinguish the two groups of deceased vs. non-deceased patients. In a subsequent second binary logistic regression with sole categorial variables, the significant predictors were re-examined to determine independent positive predictive value. Both regressions used the “Enter” method to examine the significant impact of all variables simultaneously. Based on the results, a clinical risk assessment score was developed with a relative weight depending on the regression weights. As validation cohort, a 25% sample of the existing cohort was defined by a random generator.

## 3. Results

### 3.1. Demographic and Disease Characteristics of the Study Cohort

At admission, the mean age of patients was 66 years, ranging from 19 to 89 years, and the majority were men (64%). In 253 patients (83%), one or more comorbidities were present. Thirty-one patients (10%) had concomitant endocarditis.

Routine blood tests including hemoglobin (Hb), C-reactive protein (CRP), leucocyte (Leu) and creatine (Crea) showed mean values of 11 ± 2 g/dL, 119 ± 100 mg/L, 11 ± 6 t/µL and 1.3 ± 1.3 µmol/L, respectively, and were used to screen for a reduced state of health at admission. Procalcitonin is progressively utilized as biomarker of inflammatory activity, and it was measured in 85 patients (28%) at admission revealing levels of 5 ± 18 ng/mL. Using MRI imaging, epidural and psoas abscesses were detected in 163 patients (53%) and 82 patients (27%), respectively. In 81 cases (26%) more than one anatomical location was affected by S. Lumbar region including the lumbosacral junction (L5/S1) was affected most often (n = 165, 54%). In 35 cases (11%) S manifested within the cervical/cervico-thoracic (C7/Th1) spine (Figure 1). Thoracic/thoraco-lumbar (Th12/L1) spine showed infection in 78 cases (25%, Figure 2). S affected more than two segments in a small sub-set of 36 patients (12%). In 145/307 patients (47%) a pathogen could be determined by blood cultures. In 64% of cases (n = 213) diagnostic biopsy or intraoperative samples allowed proof of pathogen. Table 1 depicts further demographic and disease characteristics of the SC and DC.

### 3.2. Course of Treatment

Overall, 185 patients (60%) did not present with a previous/preexisting antibiotic therapy. A total of 289 patients (94%) underwent invasive diagnostic transpedicular biopsy or spine surgical intervention during the course of hospitalization. Diagnostic biopsy without further spine intervention was regarded as conservative treatment pathway (n = 17, 5.5%). Overall, 35 patients (11%) were treated conservatively.

Extensive, highly specialized surgical approaches were used in 94 cases (31%). Extensive procedure includes dorso-ventral and latero-ventral approaches with autologous bone or vertebral body replacement as well as dorsal vertebral column resection. In cervical cases, anterior cervical discectomy and fusion (autologous bone or metal cage) combined with dorsal stabilization of the cervical spine was carried out (Figure 1). The majority of patients received intermediate approaches (n = 149, 49%) such as dorsal instrumentation with or without transforaminal lumbar interbody fusion/oblique lumbar interbody fusion or, at the cervical level, anterior cervical discectomy and fusion (autologous bone or metal cage) with additional ventral plating. Minimal invasive decompression surgery or abscess cleavage alone was regarded as low invasive procedure (n = 29, 9%). No patient died directly related to a spine surgical intervention. Our cohort showed an unplanned revision rate of 19% (58 cases) frequently involving dorsal wound revision or second looks (28/58).

### 3.3. Complications and In-Hospital Mortality

44 patients (14%) died while treated for spondylodiscitis. Death occurred after 22 days in mean. Patients in the deceased cohort (DC) received a mean of 13 days of intensive care medical treatment (*p* = 0.001).

Whilst routine chemical laboratory findings of surviving patients showed improvement of laboratory parameters over time, clinical parameters of patients in the MG were significantly poorer and worsening, confirming the deteriorating health as seen in Table 1.

Rheumatoid arthritis (RA) as a comorbidity was present in 22 cases (7%). All patients with RA were previously identified by a rheumatologist using the 2010 American College of Rheumatology (ACR)/European League Against Rheumatism (EULAR) classification criteria [23]. Nine of forty-four cases (20%) of RA were recorded in the DC, whilst significantly fewer cases of RA (13/250, 5%) were present in the survival cohort (SC) (*p* < 0.001). Fourteen of twenty-two patients (64%) with RA received immunosuppressive therapy, either corticosteroids alone and/or a specific rheumatic disease modifying drug or biologic response modifier [24]. Four of fourteen patients (29%) receiving immunosuppressive treatment in the RA group died.

Forty-six patients (15%) presented with preexisting neurological deficits. No independent prognostic value regarding mortality was shown for sex (*p* = 1.0), endocarditis (*p* = 0.417), neurologic deficits (*p* = 0.715), obesity (*p* = 0.75), immune suppressive therapy (*p* = 0.484) or diabetes mellitus (*p* = 0.075). Invasiveness of operative approach showed no significant impact on hospital mortality (*p* = 0.083) nor did unplanned revisions (*p* = 0.680).

Complications included acute kidney failure (n = 84, 27%), sepsis (n = 76, 25%), surgical revision (n = 58, 19%), delirium (n = 43, 14%), pneumonia (n = 41, 13%) and cardiac decompensation sepsis (n = 36, 12%). Furthermore, less than 4% of cases presented with new onset atrial fibrillation, acute liver failure, myocardial infarction, stroke, deep-vein-thrombosis or pulmonary embolism. Septic complications were the most common cause of death in our patient cohort (37/44 cases). Other causes included hemorrhagic shock (acute upper gastrointestinal bleeding), disseminated intravascular coagulopathy, acute cardiac or renal decompensation, and pneumonia. Mortality showed a very strong correlation with new onset septic conditions (*p* < 0.001).

At time of death, 39 patients (89%) received specialized multidisciplinary treatment at our intensive care unit. Five patients (11%) wished for best supportive care due to deteriorating health status, and four patients (9%) were transferred to our highly specialized palliative care unit. Table 1 depicts further demographic and disease characteristics of the SC and DC.

### 3.4. Development and Validation of a Clinical Risk Score for Mortality in S Patients

A total of 307 patients were included as the derivation cohort. The first regression model was statistically significant (χ^2^ (16) = 79.55, *p* < 0.001) and identified four independent predictors associated with in-hospital mortality in patients with S, as shown in Table 2.

The model explained 44.4% (Nagelkerke R^2^) of the variance and correctly classified 85.9% of cases. The results indicate that in case of increasing age, Crea and CRP or the presence of RA as comorbidity, the odds of in-hospital mortality significantly increased. Though marginally not significant, methicillin-sensitive *Staphylococcus aureus* (MSSA) as a causative pathogen also increased the odds for an adverse outcome.

After identifying these variables as associated risk factors, we aimed to develop a predictive score regarding in-hospital mortality in patients with S. For this purpose, the optimal threshold was calculated based on an operating characteristic curve and Youden’s J statistic. The identified thresholds associated with increased in-hospital mortality for age, Crea and CRP were 72.5 years, 1.29 mg/dL and 140.5 mg/L, respectively (Figure 3).

A rerun regression analysis with these binary variables also showed a good statistical fit. The model was statistically significant (χ^2^ (5) = 61.89, *p* < 0.001), explained 35.6% (Nagelkerke R^2^) of the variance and correctly classified 85.9% of cases, as shown in Table 3.

In a last step the significant variables age (>72.5 = 2 points), RD (3 points), creatinine (>1.29 = 2 points) and CRP (>140.5 = 2 points) were included in the development of a weighted risk score. An additional score point can be awarded when MSSA is the causative pathogen (+1 point). For practical application in practice the score is divided into four risk categories: low risk (0 points), moderate risk (1–3 points), high risk (4–6 points) and very high risk (7–10 points). Using the complete dataset, this model achieved a good predictive value for the occurrence of mortality (Table 4; area under the curve (AUC): 0.795; 95% CI, 0.719–0.871; *p* < 0.001).

To validate the score, 75 (25%) patients were randomly selected for a validation cohort that showed equivalent average key points regarding demographic and disease characteristics as well as a comparable mortality rate. The results demonstrate that, also in this sample, the risk score discriminates between the four risk groups (χ^2^ (3) = 14.15, *p* = 0.003).

### 3.5. Limitations

The single-center design limits the reliability in different settings as found in non-tertiary level hospitals. To strengthen the relevance, a separate validation cohort could be used to expand the scope of application in the future. With a short follow-up period, mortality rates in this cohort reflect only the first period of recovery and are not transferable to predict long-term outcome. Furthermore, it should be considered that there might be further clinically relevant confounding factors impacting mortality that are not accessible in our study collective. As our cohort is predominantly Caucasian, we were unable to derive further information about specific ethnic differences.

## 4. Discussion

The main findings from our study in a large cohort of 307 patients diagnosed with S are as follows: (A) increasing age, crea, CRP, preexisting RA and MSSA as causative pathogen were identified as independent predictors associated with in-hospital all-cause mortality; (B) a risk score using readily available individual and clinical characteristics showed a good predictive value with an AUC of 0.795, classifying patients in low, medium, high or very high risk groups for in-hospital mortality.

Overall, in-hospital mortality was 14.3% (n = 44) in our cohort, which is in line with other studies concerned with short-term mortality [9,10,11].

Treatment strategies for S are still controversially discussed [25,26,27]. Though many cases of S might be controllable through conservative treatment, our collective shows a relatively high rate of invasive diagnostic or spinal surgical treatment compared with the literature [25,26]. Nickerson et al., 2016 described a rate of diagnosis from percutaneous biopsies for microbiology in S varies in different studies between 14 and 76% [21]. Carragee et al., 1997 (n = 111, 60 years, multicenter, Stanford, CA, USA) reported an initial surgical rate of 40% with 38 secondary cases (34%) in which non-operative procedure failed within a month [26]. In Pola’s study, 44% of patients were treated surgically, and another 22% of conservatively treated patients received surgical biopsies [25]. In a recent study by Blecher et al., 2021 within a retrospectively analyzed conservative treatment cohort of 42 patients, 10 patients (24%) needed further instrumentation surgery due to instability [27]. In contrast to image-guided mostly fine-needle biopsies proposed by some groups, we perform transpedicular biopsies which allow better sampling due to a wider diameter. In addition, experienced spine surgeons and haptic feedback might further improve sampling, resulting in a diagnostic rate of 64% in our patient collective [28].

We trace higher surgical rates back to specific cohort factors: (A) our patients showed a high rate of epidural (53%) and psoas (27%) abscesses, which are known to lead to further disease progression in up to 49%, even after initial conservative treatment, therefore possibly requiring re-evaluation and eventually surgical procedure [22]. Abscess formation in S is also usually a sign of advanced disease stage, indicating a delayed diagnosis that corelates with a rather complicative course of disease and possible decision to proceed with surgical treatment. (B) Being a tertiary referral center for spine patients, a significant proportion of our collective (51%) already underwent treatment efforts, which to some degree led to complications that made referral necessary. Besides referrals, serious and critically ill patients are more frequently submitted directly to our emergency unit. Our department shows a high case mix index, which calculates overall lengths of stay and severity of illness, reflecting the relative urgency and complexity of patients treated in our hospital. In conclusion, our study cohort reflects a typical, contemporary S cohort at a tertiary spine center.

Several previous studies introduced risk factors of spondylodiscitis that did not show independent association with mortality in our patient cohort: (1) diabetes mellitus [2,3] with or without insulin dependent therapy (58 patients; *p* = 0.0479); (2) immunosuppressants including steroids (44 patients; *p* = 0.484), which might be due to rapidly improving treatment concepts and a multidisciplinary treatment approach.

After admission or referral to the emergency unit, the assessment of patient status is of the upmost importance to determine proper diagnostic steps and the following treatment. Scores for sepsis have long been used to improve a patient’s assessment and treatment outcomes as an essential tool in modern clinical practice [29]. We therefore highlight the need for fitting tools to be implemented to further improve the level of care when S is diagnosed.

We shall briefly discuss the results and implications for each identified risk factor of our novel HSAS score.

Age: In line with the literature, elderly patients with S are at higher risk due to their age. Akiyama et al. found age-specific graduation of mortality resulting in an increase by 2.8-fold (60–69 years), 4-fold (70–67 years) and 7.1-fold (>80 years), respectively. Hutchinson et al. reported an overall mortality of 27% in a patient cohort with a mean age of 77.5 years, which is higher than in younger cohorts [6,9,11,30]. In our study we detected a threshold > 72.5 years, which increases the odds of mortality by 3.9. Next to a higher prevalence of comorbidities, patients often present with preexisting musculoskeletal problems such as back and/or neck pain, which disguises early symptoms of S. However, in our study, age and comorbidities showed only a weak correlation (r = 0.134), putting diagnosis delays into the spotlight. In addition, the immune system weakens with age, making an older population more prone to a more rapid clinical deterioration and consequently worsened outcome [9]. With this in mind, advanced age seems to be a good indicator for an enhanced vulnerability, which can be translated into higher risk.

Rheumatoid arthritis: RA has been described as a predisposing factor for S [10]. We found a high mortality rate of 41% in this patient cohort (n = 9/22). Of particular interest, this higher mortality cannot be explained by the intake of disease-modifying or immunosuppressive medication (4/14 patients died). Recent studies showed similar results without correlation of RA medication and *S. aureus* or undefined sepsis in patients with RA [31]. This is especially important because patients usually die not because of S itself but due to septic complications and multiorgan failure. Kessler et al. identified RA as independently associated with an increased in-hospital sepsis mortality. Interpretation of reasons for this higher susceptibility to infection included high disease activity, a prematurely aged immune system (immunosenescence) and a potentially impaired humoral immunity to some pathogens [31]. RA is known to cause significant degenerative changes of the spine, masking vertebral destruction caused by S, which leads to an even further delay in diagnosis, since history of back pain in this group is often common [10]. In our cohort, 12 patients (55%) with RA had proof of an epidural abscess, of which 7 died (58%). This correlation is significantly higher than in the population without RA. Even though no statistically significant relation between epidural abscesses and mortality was found in our study, other studies described poorer prognosis in patients with epidural abscesses, with less than 50% of surviving patients making a full recovery thereafter [22]. Moreover, chronic illnesses have shown to increase the risk of a more rapid epidural abscess extension, increasing the risk of neurological deficits and complications even further [19]. In line, 23% of our patients with RD already presented with preexisting neurological deficit at admission compared to 16% in the control group. Furthermore, Stangenberg et al. were able to show that patients with RA more frequently present with disseminated manifestation of S. In their cohort, mortality broken down by localization was reported as 12% cervical, 13% thoracic, 13% lumbar and 22% for disseminated cases. Thus, this was only a trend; disseminated cases resulted twice as often in extensive, more risky surgery including ventro-dorsal approaches (44%) and subsequently higher disease burden in the RA patient collective [32].

Creatinine: In our patient cohort, creatinine levels of >1.29 mg/dL led to an increase in mortality by 4.3-fold. Overall, 65 patients showed some degree of chronic renal dysfunction, of which 21/44 (48%) were found in the DC and only 23/263 (9%) in the SC (*p* < 0.001). Management of patients with end-stage renal disease, especially when hemodialysis is needed, has proven challenging with odds for mortality increased by up to 10-fold [2]. It has previously been shown that, besides immune dysfunction associated with uremia, patients with severe renal dysfunction have been shown to carry a higher prevalence of co-morbidities such as chronic heart disease, further enhancing an already high vulnerability to infection [33]. Furthermore, an increased risk of Staphylococcal infection and bacteremia, partly due to repetitive vascular puncture and instrumentations, has been shown in this cohort [33,34].

CRP: The Infectious Diseases Society of America attributes a strong recommendation to consider S in patients presenting with new or worsening back or neck pain and elevated CRP in their Clinical Practice Guidelines [6]. In our cohort, patients in the DC showed especially high CRP values (178 mg/dL) at admission. Mortality in S results not merely from the spine as infection focus itself but in most cases because of the systemic complications that follow [33]. Septic distribution through spread into the blood stream (bacteremia) or cerebrospinal fluid (meningitis) can have severe consequences resulting in multiorgan failure and shock, where CRP levels are known to be high. Furthermore, it is known that CRP in septic patients carries significant independent prognostic information [35]. In our cohort, 100 patients (33%) showed signs of sepsis at administration, and 76 (25%) patients developed such complication during their hospital stay. The Sepsis-3 criteria introduced in 2016 describe sepsis as a life-threatening immune response to infection [29]. Considering existing sepsis guidelines, any kind of organ dysfunction in patients with S must be considered a red flag [35].

MSSA: When MSSA was detected in patients presenting with septic conditions, an alarmingly high mortality rate of 70% (41/59 cases; *p* < 0.001) was registered. Kehrer et al. reported mortality rate ratios of 8.8 for up to 90 days and 1.4 for up to one year in cases with unknown etiology in contrary to 24 for up to 90 days and 6 for up to one year among *S. aureus* infections [11]. Furthermore, another study found that 60% of all cases with *S. aureus* detection had (or progressed to developing) an epidural, paraspinous or psoas abscess [36]. Although in our study the 2.3 increased odds were just barely not statistically significant, combined with recent literature we believe an additional risk-point should be awarded indicating higher risk of a fast, more aggressive disease progression or failure of therapy [37].

## 5. Conclusions

With the help of our novel spondylodiscitis assessment score, the treating team of physicians is enabled to identify patients at high risk of a fatal course of spondylodiscitis upon admission to the hospital. The score can be fully ascertained after a short time and could influence diagnostic decisions and accelerate time to effective treatment. The score might therefore contribute to an improved outcome, particularly in vulnerable patients.

## Figures and Tables

**Figure 1 jcm-11-00660-f001:**
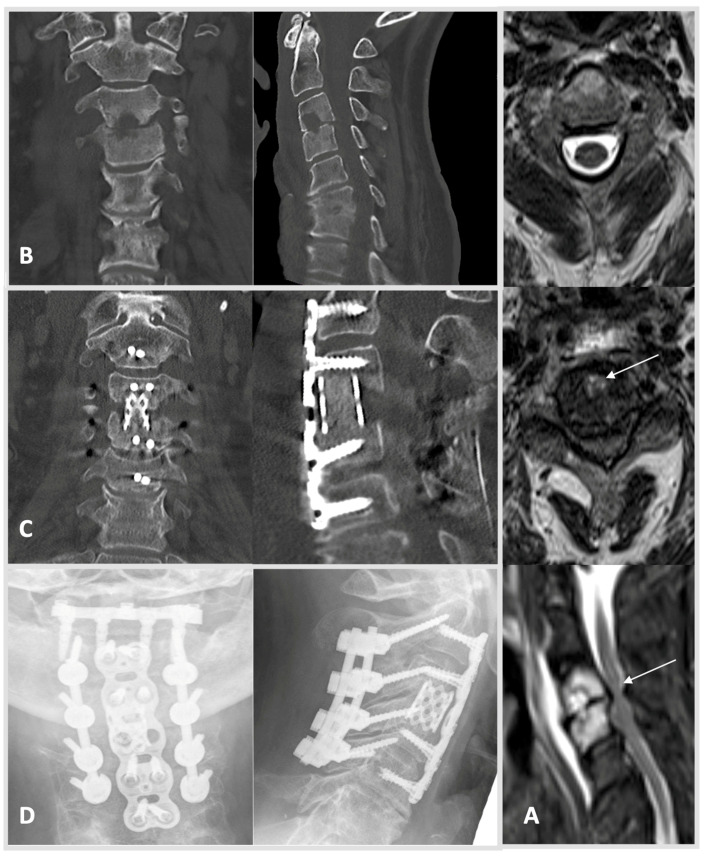
Case 1: A 61-year-old male patient was admitted after showing severe neurologic deficits at a follow-up appointment. The patient had been receiving radiotherapy after left-sided lung resection due to non-small-cell lung cancer, which was diagnosed 2015. He presented with incomplete tetraparesis with 3–4/5 degrees of strength hypesthesia in both hands (AIS scale D). CRP was 75 mg/L, and creatinine was measured at 0.97 mg/dL. HSAS risk score: low risk. (**A**) MRI scans showed a large retropharyngeal abscess, spondylodiscitis of vertebrae C3/4, and cervical myelopathy. MRI from top to bottom: free intraspinal conditions cranially (C2 axial plane, T2 weighted); severe stenosis with concomitant hyperintense myelopathy signal (arrow, C3/4 axial plane, T2 weighted); sagittal plane after contrast agent with large prevertebral abscess, relevant contrast enrichment of C3 and 4 and subtotal stenosis C3/4 with hyperintense myelopathy signal (arrow). (**B**) CT scans of the whole spine showed concomitant bony destruction within the segment C3/4 (left: coronal, right: lateral). (**C**) Navigated dorsal instrumentation C2–5 with laminectomy C3 and 4 and subsequent spondylodesis with autologous bone was performed for extended decompression. In a ventral approach, partial vertebrectomy C3 and 4 with debridement and interbody fusion using a spongious bone filled Harms-cage supported ventrally by an anterior cervical plate C2–5. (**D**) Radiographs in coronal and lateral views show a good result four months postoperatively. Intraoperative samples showed *staphylococcus epidermidis*. After anti-infective treatment and rehabilitation, the patient regained a strength level at 4–5/5 degrees and good function with persistent but compensated gait ataxia at follow-up four months postoperatively.

**Figure 2 jcm-11-00660-f002:**
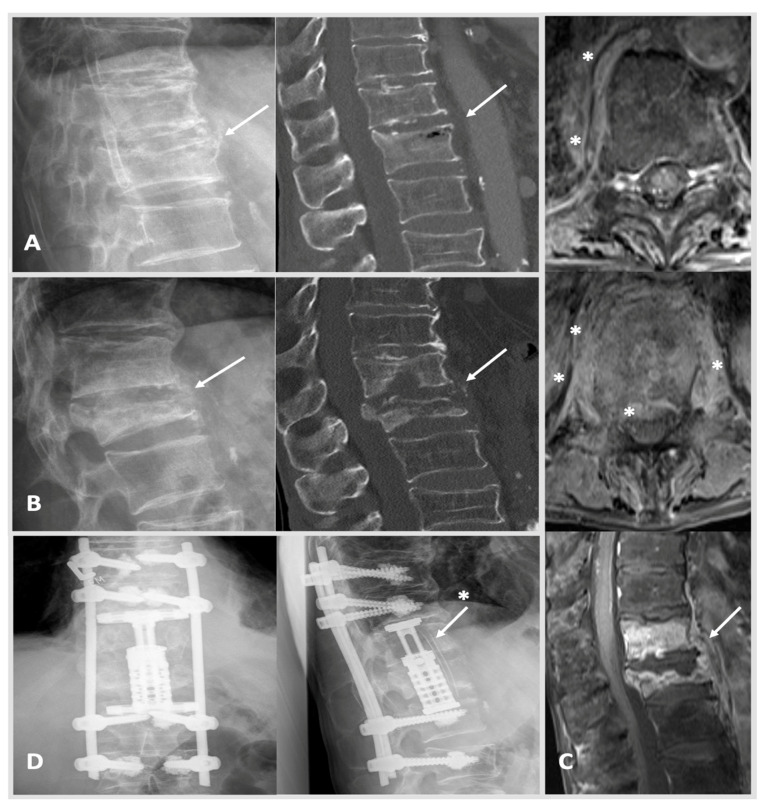
Case 2: A 66-year-old male patient was treated due to a worsening health status and proven bacteremia (*staphylococcus epidermidis* and multiresistant *enterococcus faecium*) of unknown origin. A thoracic CT showed a secondary, suspectedly older, fracture of Th 12 type AO Spine A3 (image (**A**): left: X-ray; right: CT scan with intravertebral vacuum phenomenon; arrow marking Th 12). MRI imaging was initiated 6 weeks after the image (**A**) when yet another thoracic CT showed progressing destruction and spondylodiscitis of Th 12 was diagnosed. Initially, CRP was measured at 30 mg/L, and creatinine was documented at 0.99 mg/dL. HSAS risk score: low risk. The patient requested a conservative treatment and agreed to spine surgical transpedicular biopsy, which confirmed the previously identified pathogen *S. epidermidis*. As primary infectious foci, the patient’s port-catheter was identified and removed promptly, which was in place for a previous esophageal cancer. The patient received anti-infective treatment and returned for follow-up at our out-patient clinic ten weeks later. Here, CRP rose to 100 mg/L, and the patient reported worsening backpain. MRI and CT scans were conducted as well as X-rays standing up showing massive bony destruction of Th 12 and also Th 11 with kyphosis. Image (**B**): left, X-ray with progressing kyphosis when standing upright; right, CT scan showing pathological fracturing of Th 12 (arrow) type AO Spine A4. Image (**C**): MRI from top to bottom: free intraspinal conditions cranially with Th 10 surrounding prevertebral tissue reaction marked with stars (axial plane, T1 weighted); more distinctive prevertebral and epidural tissue reaction marked with stars (Th 12 axial plane, T1 weighted); large prevertebral abscess Th10–L1 (arrow) with strong enrichment of Th 11 and Th12 (T1 weighted). Image (**D**): in a two-step procedure (1) dorsal cement-augmented percutaneous stabilization of Th9–L2 and (2) a lateral minimal invasive thoracotomy approach with incomplete resection of costa 10 was used to achieve anterior stabilization via partial vertebrectomy Th 11 and 12 with debridement and vertebral body replacement with concomitant placement of autologous bone stock i.e., rib (image (**D**): arrow with star) counteracting kyphosis was established (postoperative X-rays: left anterior-posterior, right lateral view). The patient received anti-infective treatment, and after a long hospitalization, he was able to return to his nursing home. Last follow-up showing after 12 months showed stable clinical findings.

**Figure 3 jcm-11-00660-f003:**
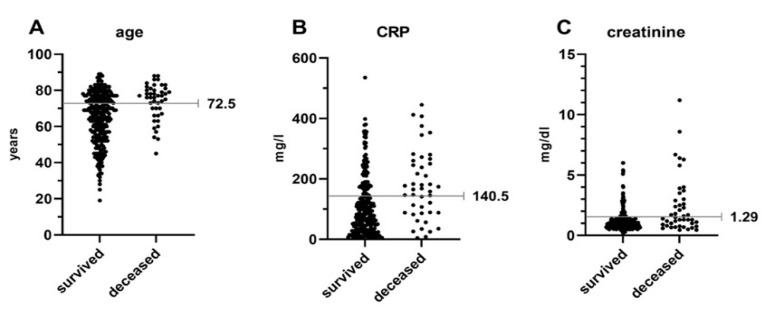
Optimal thresholds were defined using an operating characteristic curve and Youden’s J statistic. Points depict individual values for patients of the survival and deceased group separately for age (**A**), CRP (**B**) and creatinine (**C**). CRP = C-reactive protein.

**Table 1 jcm-11-00660-t001:** Selection of further demographic and disease characteristics of the survival cohort and deceased cohort in means of *p*-values.

	Survival Group Mean (SD) or n (%)	Mortality Group Mean (SD) or n (%)	*p*-Value
Age	64.4 (14.6)	73.8 (9.9)	<0.001
Body mass index	28 (19.6)	26.9 (7.5)	0.75
ICU stay (in days)	7.1 (19.5)	13.3 (11.1)	<0.001
Transfusion (intra- and postoperative)	1.2 (2.9)	2.3 (3)	0.14
Creatinine preoperative	1.2 (.9)	2.4 (2.5)	<0.001
Last Creatinine measurement	1.1 (.9)	1.8 (1)	<0.001
Hb preoperative	10.9 (1.9)	10.1 (1.8)	0.01
Last Hb measurement	9.5 (1.5)	8.4 (1)	<0.001
PCT preoperative	1.1 (2)	15.1 (33)	<0.001
Last PCT measurement	0.4 (.6)	15.9 (50)	<0.001
Leucocytes preoperative	10.1 (5.8)	12.8 (5.5)	0.003
Last Leucocyte measurement	7 (2.6)	19.7 (14.3)	<0.001
Chronic heart disease	35/50 (70)	15/50 (30)	0.001
Chronic kidney failure	44/65 (67.7)	21/65 (32.3)	<0.001
Rheumatoid arthritis	13/22 (59)	9/22 (41)	<0.001
Acute kidney failure	55/84 (65.5)	29/84 (34.5)	<0.001
Acute cardiac decompensation	23/36 (63.9)	13/36 (36)	<0.001
Acute liver failure	2/9 (22)	7/9 (77.8)	<0.001
Pneumonia	30/41 (73)	11/41 (26.8)	0.028

ICU = Intensive care unit; Hb = hemoglobin; PCT = procalcitonin; SD = standard deviation.

**Table 2 jcm-11-00660-t002:** Logistic regression for independent predictors of mortality in patients with spondylodiscitis. Regression with “Enter” method.

							95% CI	
Predictor	β	SE β	Wald’s χ^2^	*df*	*p*-Value	OR	Lower	Upper
Constant	−7.950	2.203	10.098	1	<0.001	0.001	NA	NA
**Age**	**0.067**	**0.023**	**8.702**	**1**	**0.003**	**1.069**	**1.023**	**1.118**
Heart failure	0.796	0.505	2.483	1	0.115	2.217	0.824	5.967
CKD	0.440	0.484	0.824	1	0.364	1.552	0.601	4.013
**RhA**	**2.338**	**0.687**	**11.581**	**1**	**0.001**	**10.360**	**2.695**	**39.827**
Endocarditis	0.195	0.656	0.088	1	0.767	1.215	0.336	4.395
** *S. aureus* **	**1.085**	**0.537**	**4.078**	**1**	**0.043**	**2.959**	**1.032**	**8.481**
CNS	0.624	0.690	0.817	1	0.366	1.866	0.482	7.221
Enterobacterales	−0.691	1.313	0.277	1	0.599	0.501	0.038	6.572
*Enterococcus* sp.	0.975	0.943	1.069	1	0.301	2.651	0.418	16.821
*Strep.* sp.	0.558	1.175	0.225	1	0.635	1.746	0.175	17.471
*M. tuberculosis*	−13.97	40,292	<0.001	1	1.0	0.000	0.000	-
Other germ sp.	1.893	1.112	2.902	1	0.088	6.642	0.752	58.684
Hemoglobin	−0.230	0.128	3.217	1	0.073	0.795	0.618	1.022
Leukocyte count	0.015	0.038	0.165	1	0.685	1.016	0.943	1.094
**Creatinine**	**0.462**	**0.144**	**10.263**	**1**	**0.001**	**1.587**	**1.196**	**2.105**
**CRP**	**0.006**	**0.002**	**7.494**	**1**	**0.006**	**1.006**	**1.002**	**1.010**
Test			χ^2^	*df*	*p*-value			
Overall model evaluation Omnibus test			79.55	16	<0.001			
Goodness-of-fit test Hosmer–Lemeshow test			6.2043	8	0.624			

Cox and Snell R^2^ = 0.263; Nagelkerkes R^2^ = 0.444. CI = confidence interval; RhA = Rheumatoid arthritis; CKD = Chronic kidney disease; CRP = C-reactive protein; CNS = Coagulase-negative Staphylococci; *Strep.* = Streptococcus; sp. = species; *df* = degrees of freedom; OR = odds ratio, SE = standard error. Bold is the independent risk factor used within the proposed score.

**Table 3 jcm-11-00660-t003:** Dichotomous variables as independent predictors of mortality in patients with spondylodiscitis. Logistic regression with “Enter” method.

		95% CI		
	Odds Ratio	Lower	Upper	*p*-Value
Age				
>72.5	3.86	1.72	8.67	0.001
≤72.5	1			
Rheumatoid arthritis				
yes	9.37	2.63	33.35	0.001
no	1			
*S. aureus* infection				
yes	2.27	1.0	5.16	0.051
no	1			
Creatinine				
>1.29	4.35	1.95	9.68	<0.001
≤1.29	1			
CRP				
>140.5	4.07	1.83	9.02	0.001
≤140.5	1			

CI = confidence interval; *S.* = *Staphylococcus*; CRP = C-reactive protein; Cox and Snell R^2^ = 0.210; Nagelkerkes R^2^ = 0.356; correctly classified 85.9%.

**Table 4 jcm-11-00660-t004:** Clinical classification of the risk score for estimating the risk of mortality in patients with spondylodiscitis (n = 307). Area under the curve (AUC): 0.795; 95% CI, 0.719–0.871; *p* < 0.001.

	Deceased (n)	Survived (n)	Total (n)	*p*-Value (χ^2^)
Low risk (0 points)	1	87	88	<0.001
Moderate risk (1–3 points)	8	113	121	
High risk (4–6 points)	27	60	87	
Very high risk (7–10 points)	8	3	11	

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
