# Peer review of "The Hamburg Spondylodiscitis Assessment Score (HSAS) for Immediate Evaluation of Mortality Risk on Hospital Admission"

_jcm, 2022, doi:10.3390/jcm11030660_

Round 1

Reviewer 1 Report

Authors present a retrospective study on 307 patients with spondylodiscitis in order to develop a risk assessment score - Hamburg Spondylodiscitis Assessment Score (HSAS) - using fast available parameters to predict in-hospital mortality. Mortality rate was 14%;  age older than 72.5 years, rheumatoid arthritis, creatinine >1.29 mg/dl, CRP > 140.5 mg/l increased the risk of mortality 3.9-fold, 9.4- 
fold, 4.3-fold and 4.1-fold, while S. aureus detection increased the risk of mortality by 2.3- fold

The manuscript is well written, however there are several issues which need to be clarified. 94% of the patients underwent diagnostic or spine surgical therapy, then again 10% had a conservative treatment - please clarify, as the numbers do not add up - how many patients did receive a surgery and which kind of surgery was it. 

Although the primary intention was to develop a risk assesment score using fast available parameters, unfortunately one very important parameter which is fast available was not taken into consideration - neurological deficits prior to surgery. Patients with neurological deficits, i.e. inability to walk prior to surgery, will probably have higher risk of mortality. Please state the causes of mortality - pulmonary embolism, incidence. 

Please clarify why were variables such as procalcitonin, IL-6, leukocyte count, positive blood culture and neurological deficits taken into the score. 

It is surprising that the details of the surgical therapy - stabilization with/without cage, simple decompression and empyema evacuation. These parameters should be reported and taken into the evaluation of the in-hospital mortality as a separate variable - type of surgery, duration of surgery, blood loss, possible operative complications - could all have a decisive impact on the patients prognosis. Please state the complications - which were they, complication rate, how many wound revision, CSF leaks, screw malposition if applicable.

Furthermore, localization of the spondylodiscitis could also play a potential role for in-patient mortality - please report on localization. 

Did you find any gender- or racial differences?

Please provide several illustrative cases of spondylodiscitis in the cervical, lumbar and thoracic spine. 

For Discussion I suggest to include following studies and comment:

  1. Pojskić, M.; Carl, B.;Schmöckel, V.; Völlger, B.;   Nimsky, C.; Saß, B. Neurosurgical Management and Outcome Parameters in 237 Patients with
    Spondylodiscitis. Brain Sci. 2021, 11,1019. https://doi.org/10.3390/ brainsci1108101 - please comment on screening for concomitant infections; did you perform it and did you have a protocoll for this; did you screen for other infections beside endocarditis
  2. Mazarakis NK, Baren J, Loughenbury PR, Koutsarnakis C, Gupta H, Fawcett RW. Site matters: Image-guided percutaneous sampling of intervertebral disc results in increased positive diagnostic yield in spondylodiscitis. Br J Neurosurg. 2021 Dec 14:1-5. doi: 10.1080/02688697.2021.2013438. Epub ahead of print. PMID: 34904496. - did you perform sampling from the disc and how many samples per surgery
  3. Blecher R, Frieler S, Qutteineh B, Pierre CA, Yilmaz E, Ishak B, Glinski AV, Oskouian RJ, Kramer M, Drexler M, Chapman JR. Who Needs Surgical Stabilization for Pyogenic Spondylodiscitis? Retrospective Analysis of Non-Surgically Treated Patients. Global Spine J. 2021 Sep 16:21925682211039498. doi: 10.1177/21925682211039498. Epub ahead of print. PMID: 34530628. - which criteria do the authors in their own department take as a standard for stabilization 

Reviewer 2 Report

The authors described development of the risk-evaluating scoring system for mortality based on retrospactive cohort of the patients with pyogenic spondylitis. Data presented here is well convincing and the scoring system will be effective in clinics. There are minor points to be considered. 

1) Did you analyze diabetus as one of well-known risk factors for mortality of deterioration of pyogenic spondylitis?

2) Same as immunosuppressants including steroids.

Round 2

Reviewer 1 Report

The authors have presented an extensive revision with significant improvement of their retrospective study on mixed cohort of conservative and surgical treatment of 307 patients with spondylodiscitis, which led to the establishment of the The Hamburg Spondylodiscitis Assessment Score (HSAS) for immediate evaluation of mortality risk on hospital admission. I congratulate the authors for hard quality work on the first and second version of the manuscript and sincerely hope that HSAS will find its way in a broad clinical practice and a place in the further academic analysis of vertebral osteomyelitis.